# Parkinson's disease and cancer: a systematic review and meta-analysis of over 17 million participants

Xinyuan Zhang,[1] David Guarin ,[2] Niyaz Mohammadzadehhonarvar,[2] Xiqun Chen,[2] Xiang Gao [1]

► Prepublication history and additional online supplemental material for this paper are available online. To view these files, please visit the journal online (http://dx.doi.org/10.1136/bmjopen-2020-046329).

XZ and DG are joint first authors. XC and XG are joint senior authors.

[1]Nutritional Science, The Pennsylvania State University, University Park, Pennsylvania, USA
[2]Neurology, Massachusetts General Hospital, Boston, Massachusetts, USA

**Correspondence to**
Dr Xiqun Chen;
XCHEN17@mgh.harvard.edu

## ABSTRACT

**Objective** To systematically review and qualitatively evaluate epidemiological evidence on associations between Parkinson's disease (PD) and cancer via meta-analysis.

**Data sources** MEDLINE via PubMed, Web of Science and EMBASE, until March 2021.

**Study selection** Included were publications that (1) were original epidemiological studies on PD and cancer; (2) reported risk estimates; (3) were in English. Exclusion criteria included: (1) review/comments; (2) biological studies; (3) case report/autopsy studies; (4) irrelevant exposure/outcome; (5) treated cases; (6) no measure of risk estimates; (7) no confidence intervals/exact p values and (8) duplicates.

**Data extraction and synthesis** PRISMA and MOOSE guidelines were followed in data extraction. Two-step screening was performed by two authors blinded to each other. A random-effects model was used to calculate pooled relative risk (RR).

**Main outcomes and measures** We included publications that assessed the risk of PD in individuals with vs without cancer and the risk of cancer in individuals with vs without PD.

**Results** A total of 63 studies and 17 994 584 participants were included. Meta-analysis generated a pooled RR of 0.82 (n=33; 95% CI 0.76 to 0.88; p<0.001) for association between PD and total cancer, 0.76 (n=21; 95% CI 0.67 to 0.85; p<0.001) for PD and smoking-related cancer and 0.92 (n=19; 95% CI 0.84 to 0.99; p=0.03) for non-smoking-related cancer. PD was associated with an increased risk of melanoma (n=29; pooled RR=1.75; 95% CI 1.43 to 2.14; p<0.001) but not for other skin cancers (n=17; pooled RR=0.90; 95% CI 0.60 to 1.34; p=0.60).

**Conclusions** PD and total cancer were inversely associated. This inverse association persisted for both smoking-related and non-smoking-related cancers. PD was positively associated with melanoma. These results provide evidence for further investigations for possible mechanistic associations between PD and cancer.

**Prospero registration number** CRD42020162103.

## INTRODUCTION

Parkinson's disease (PD) is the second most common neurodegenerative disease affecting more than 10 milion people worldwide. It is characterised by premature cell death of dopaminergic neurons in the substantia nigra pars compacta. Clinically, PD is manifested by tremor, rigidity, bradykinesia and postural instability. Non-motor symptoms are also common. Symptomatic treatments for PD are available and effective, however, there is currently no therapy known to modify disease progression. Among enviromental factors that have been associated with the risk of developing PD, age is the main risk factor, whereas smoking has been inversely associated with PD.[1 2] Familial PD accounts for 5%–15% of total PD. The most common genetic cause of PD is mutations in *LRRK2*. Other PD-related genes include *PARK2*, *PARK7*, *PINK1* and *SNCA*. PD is increasingly recognised as a systemic disorder. Oxidative stress, mitochondria dysfuntion, energy failure, immune dysregulation and chronic inflammation have been proposed to contribute to neurodegeneration in PD.[3]

Cancer is characterised by uncontrolled cell proliferation and growth. It is among the leading causes of death worldwide.[4] Growing evidence suggests that PD and cancer may be associated.[5] Similar to PD, cancer incidence increases with age.[6] Smoking also modifies the risk of certain cancer, especially lung cancer, though in the opposite direction to the risk of PD.[7] In addition, PD-related genes have been implicated in cancer. *PARK2* has been identified as a potent tumour suppressor gene, whereas mutations in *LRRK2* have been associated with an increased risk of cancer.[8] While

## Strengths and limitations of this study

► Unlike recent meta-analyses, this study stratifies analysis for smoking vs non-smoking cancers.
► Heterogeneity between included studies was analyzed via meta-regression.
► Despite best efforts, high heterogeneity in methodology and cohorts of included studies cannot be fully dealt with by statistical methods.

a positive, bidirectional link between PD and melanoma, a malignant tumour that develops from melanocytes is well documented,[9] there appears to be an inverse association between PD and total cancer.[10] However, it remains unclear whether PD and cancer are associated mechanistically, or the findings were confounded by other factors, such as study designs and smoking. Clearly documenting these associations is important for bridging the interdisciplinary knowledge gap and developing novel preventive and treatment strategies for both PD and cancer. An individual study may lack the power to detect an association. A meta-analysis can increase precision in estimating risk,[11] especially in subsets of cancers with even fewer cases. We, thus, conducted a meta-analysis to systematically review the population-based evidence for the potential association between PD and cancer. To better elucidate PD-cancer relation, we first stratified studies according to the temporal association between the two diseases into three categories: PD preceding cancer, cancer preceding PD and co-occurrence. Second, we performed sensitivity analyses in which variations in study design and qualities, and levodopa treatment, were evaluated. Third, we separately analysed smoking-related cancers and non-smoking-related cancers to address smoking as a potential confounding factor. Finally, we specifically analysed the associations between PD and melanoma, non-melanoma skin cancers and other major cancers (eg, prostate cancer, colon cancer and breast cancer).

## METHODS

### Literature search and data extraction

This meta-analysis followed the MOOSE guidelines for reporting meta-analysis on observational studies and was registered on PROSPERO (CRD42020162103). We searched all published literature that reported PD association with cancer in MEDLINE via PubMed, Web of Science and EMBASE up to 1 March 2021. Search items related to 'PD', 'cancer' and 'epidemiological studies' were identified and modified for each database. We constrained our search in human studies and in the English language. Detailed search terms can be found in online supplemental materials. Duplicates were matched based on author, year, and title in Endnote X9 and manually compared before removing.

The inclusion criteria were: (1) original studies that were conducted in an epidemiological setting; (2) studies reported either an OR, relative risk (RR), HR, standardised incidence/mortality ratio (SIR/SMR) or other reliable measures of estimated risk; (3) studies in which PD and cancer cases were ascertained by doctor's diagnosis, hospitalisation record, disease identification codes or self-report on the diagnosis. Exclusion criteria included: (1) reviews or comments; (2) non-epidemiological studies; (3) case reports/autopsy studies; (4) irrelevant exposure/outcome; (5) treated cases; (6) no measure of risk estimates; (7) no confidence intervals/exact p values and (8) duplicates. Parkinsonism that

does not meet the criteria for PD and benign neoplasm were not included. Previous meta-analyses were used as references for manual searching of related publications. Two first authors (XZ and DG) independently screened the publications in two steps: title/abstract screening and full-text screening. Any discrepancy was reviewed and reconciled by two senior authors (XC and XG). During full-text screening, we found five groups of publications using the same population or dataset. Details of inclusion and exclusion step are reported in online supplemental methods. After screening references of included publications, we found two other eligible publications that were not captured by search items.[12 13]

### Data extraction

From each of the included publications, we extracted information on the first author, year of the study, study type, country origin, population, mean age, dominant sex, dominant ethnicity, cases and controls population size, measure of risk, PD and cancer ascertainment methods, adjusted covariates, levodopa use and estimated risk with lower and upper CIs for each type of cancer. The temporal association was defined per each individual study definition, most of which was based on the diagnosis date of the two diseases. Dominant sex and ethnicity were defined as the major sex and race/ethnicity (>50%) of the studied population, respectively.

The type of study was categorised into prospective study, case–control study, case-only cohort study and cross-sectional study.

### Statistical analysis

All analyses were performed in STATA SE V.15. Cochran's Q statistic and $I^2$ were calculated to examine heterogeneity among studies. Cochran's Q was computed as the sum of variance from the pooled estimates and compared with chi-squared distribution with $k$-1 ($k$=number of publications) degree of freedom. $I^2$ was calculated as the percentage of variation across studies due to heterogeneity rather than chance.[14] Due to the high heterogeneity of included publications (p value for Q statistics <0.05, $I^2$ >50% for all), pooled effect sizes (including RR, OR, HR, SIR and SMR) were calculated using random-effects models to account for unobserved heterogeneity. Egger test and funnel plots were performed to assess publication bias.

For total cancer, we performed three sensitivity analyses. First, four publications from meeting proceedings/abstracts were further included; second, eight mortality publications were excluded; third, two publications using invalidated, self-report diagnosis of either cancer or PD were excluded. Further, we performed six subgroup analyses, looking at the variance of the included publications in population age, dominant sex, dominant race/ethnicity, study design, study quality and year of study. Age was separated into two groups by the mean age of the included studies (69.3 years). Dominant ethnicity was categorised into Caucasian-dominant and

Asian-dominant. The study design was categorised into cohort studies and other types of studies. Study quality was assessed by the Newcastle-Ottawa Scale for cohort studies and for case–control studies,[15] which is based on the definition of case/control, the definition of exposure/outcome, covariates and other relevant factors. The score ranged from 0 to 9, and we separated the included studies into low-quality group (<7) and high-quality group (≥7), based on the mean quality score of the included studies. Proceedings/abstracts were not included in the quality check. The difference between groups was tested by the meta-regression method.

We categorised cancers into smoking-related and non-smoking-related cancers according to National Cancer Institute and Centers for Disease Control and Prevention's definition.[16] Smoking-related cancers include cancer of the lung, larynx, mouth, oesophagus, throat, bladder, kidney, liver, stomach, pancreas, colon and rectum, and cervix, as well as acute myeloid leukaemia. Cancers of other sites, including melanoma, were regarded as not associated with smoking. If a publication reported grouped smoking- and non-smoking-related cancers, the risk estimates were extracted directly. If a publication reported individual cancers only, and the number of sites is more than 10, we first categorised individual cancers into smoking-related and non-smoking-related groups accordingly,[16] calculated pooled RR and 95% CI in each group using a random-effects model, and then included the resulting pooled RR in the final meta-analysis.

We specifically evaluated the association between PD and melanoma, and other skin cancers. Cancers of other specific sites were included in this meta-analysis if there were more than 10 publications. Included were lung cancer, colorectal cancer, breast cancer and prostate cancer.

## RESULTS

In total, we included 63 publications in this meta-analysis (figure 1).[12 13 17–77] Characteristics of all publications are listed in online supplemental table 1.

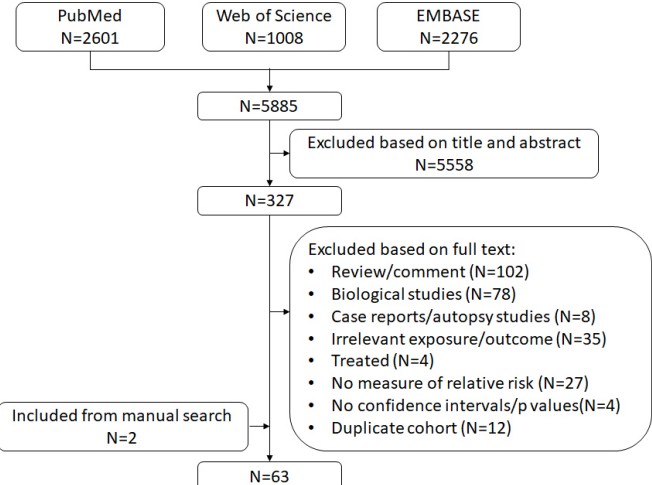

**Figure 1** Flow chart.

### PD and total cancer

Combining 33 publications,[12 13 18 27–32 35–39 41 50–52 54 57–62 64 65 69–71 73 75 76] pooled RR for association between PD and cancer was 0.82 (95% CI 0.76 to 0.88; p<0.001; figure 2). We did not observe evidence for existence of publication bias (Egger test p=0.27; online supplemental figure 1). After stratified by temporal sequence, PD was significantly associated with a lower future risk of cancer (n=21, pooled RR=0.85; 95% CI 0.76 to 0.95; p=0.004), and similar association was observed for cancer with a lower future risk of PD (n=11, pooled RR=0.74; 95% CI 0.65 to 0.85; p =<0.001). The significant inverse association persisted after further including meeting abstracts, excluding mortality studies and excluding self-report outcomes that were not validated (table 1). Meta-regression did not find significant difference between subgroups stratified by age (<69.3 years vs ≥69.3 years; mean value of the included studies), sex (men-dominant vs women-dominant cohorts), ethnicity (Caucasian vs Asian), study design (cohort vs others), study quality (scored <7 vs ≥7) or year of study (before 2010, or 2010 and after (online supplemental table 2).

We found four publications that examined the risk of cancers associated with the treatment of levodopa in PD patients (online supplemental table 3).[18 24 31 56] Although there was a significant lower risk of cancer after levodopa treatment or with higher cumulative levodopa treatment (pooled RR=0.75; 95% CI 0.61 to 0.92; p=0.007; online supplemental figure 2a), Egger test (p=0.005) and funnel plot (online supplemental figure 2b) showed a significant publication bias and thus a potentially overestimated result.

### Smoking-related and non-smoking-related cancers

Combining 21 publications,[12 18 28 30–32 35 37 38 50 51 54 57–60 66 70 71 73 76] the pooled RR for association between PD and smoking-related cancers was 0.76 (95% CI 0.67 to 0.85; p<0.001; figure 3A). PD was also inversely associated with non-smoking-related cancers (n=19; pooled RR=0.92; 95% CI 0.84 to 0.99; p=0.03; figure 3b).[12 18 25 28 30 31 35 37 38 50 51 57–59 66 70 71 73 76] No publication bias was observed for both analyses (Egger test p=0.45 and 0.50, respectively; online supplemental figure 3).

### Melanoma and non-melanoma skin cancer

Combining 29 publications,[17 18 20 23 24 26 30 32 35 37 38 43 47 50 51 54 56 57 59 60 64 67–71 73 76 77] the pooled RR for association between PD and melanoma was 1.75 (95% CI 1.43 to 2.14; p<0.001; figure 4A). No publication bias was observed (Egger test p=0.28; online supplemental figure 4a). We did not find a statistically significant association between PD and non-melanoma skin cancer (n=17; pooled RR=0.90; 95% CI 0.60 to 1.34; p=0.60; figure 4B).[31 32 34 35 37 41 47 50 54 56 57 67 69 71 73 76 77] Egger test suggested no publication bias (p=0.53), but funnel plot suggested potential overestimation by small studies (online supplemental figure 4b).

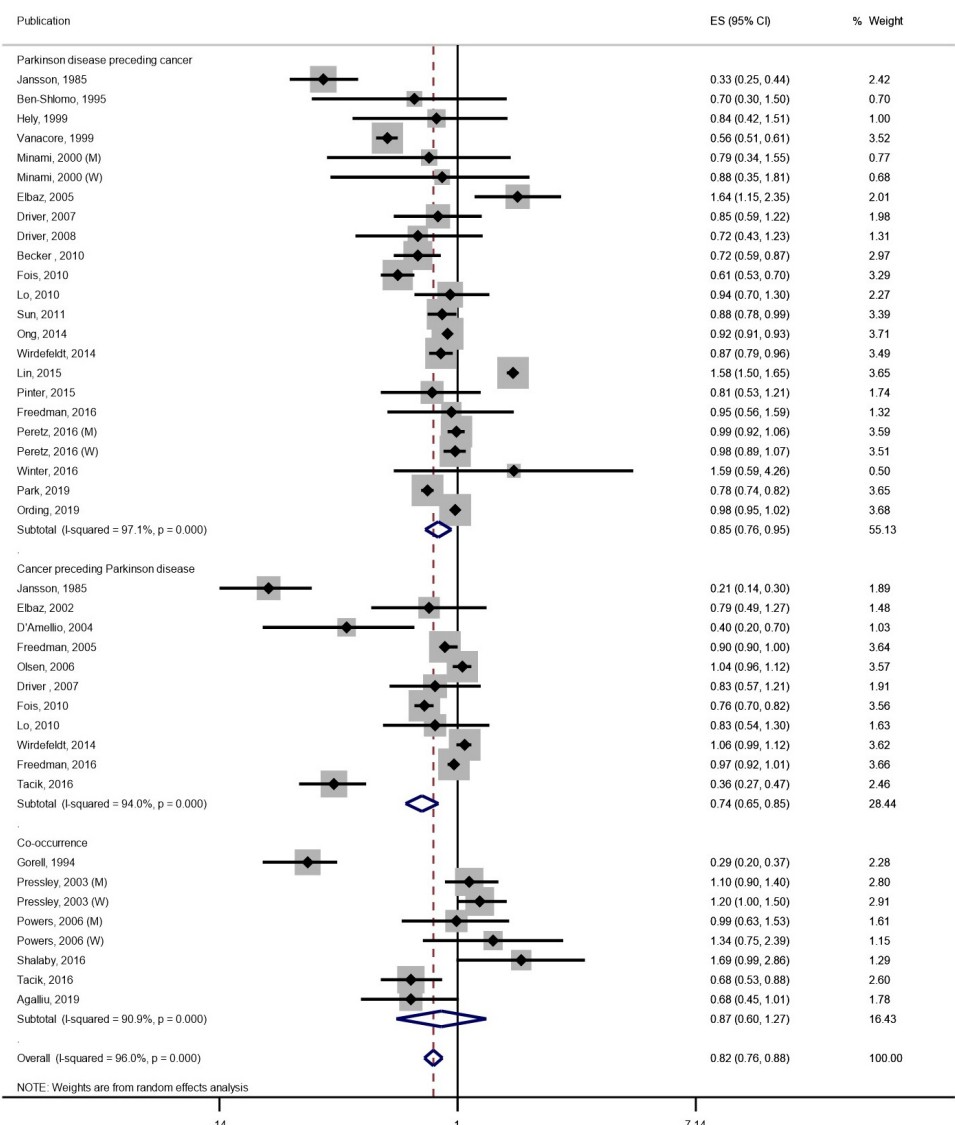

**Figure 2** Association between Parkinson's disease and total cancer in 33 publications. The figure shows the estimates (ESs) and 95% CIs for each study and the pooled result from random-effects model. Studies are stratified by temporal relationship of Parkinson's disease and cancer. M, men; W, women.

## Other site-specific cancers

Lung cancer and colorectal cancer, two major cancers in the smoking-related category, both showed a significant inverse association with PD. There was no significant association between PD and breast cancer and prostate cancer (table 1).

## DISCUSSION

In this meta-analysis of 63 publications and 17 994 584 participants, a significant inverse association between PD and total cancer was observed, with an 18% lower risk on both sides. Individuals with PD had a 15% lower risk of developing cancer, and vice versa, individuals with cancer had a 26% lower risk of developing PD. The inverse association was stronger for smoking-related cancers, compared with non-smoking-related cancers, though both achieved statistical significance. In contrast, PD was

significantly associated with a 75% higher risk of melanoma. The overall inverse association is consistent with two published meta-analyses on this topic, which reported a 27% and 6% significantly lower risk for total cancer, respectively.[10][78] Relative to these two published meta-analyses, our study included a large number of studies and participants. The latest meta-analysis, for example, included 15 studies and 1 480 239 participants for examining the association between PD and total cancer.[10][78] In addition, this study did not stratify smoking-related and non-smoking-related cancers despite the analysis of associations between PD and specific cancers. Further, these two meta-analyses included both PD and parkinsonism.[10][78]

One of the possible explanations for the inverse association between PD and total cancer is smoking. Smoking has been consistently associated with a low risk of PD and

**Table 1** Association between Parkinson's disease and cancer

| | No of publications | Pooled RR (95% CI) | P for significance | P for heterogeneity |
|---|---|---|---|---|
| Total cancer | | | | |
| All full-text publications | 33 | 0.82 (0.76 to 0.88) | <0.001 | <0.001 |
| Including abstracts | 37 | 0.80 (0.74 to 0.86) | <0.001 | <0.001 |
| Excluding mortality studies | 25 | 0.85 (0.79 to 0.92) | <0.001 | <0.001 |
| Excluding self-report diagnosis | 31 | 0.81 (0.75 to 0.87) | <0.001 | <0.001 |
| Smoking-related cancer** | 21 | 0.76 (0.67 to 0.85) | <0.001 | <0.001 |
| Non-smoking-related cancer†† | 19 | 0.92 (0.84 to 0.99) | 0.03 | <0.001 |
| Site-specific cancer | | | | |
| Melanoma | 29 | 1.75 (1.43 to 2.14) | <0.001 | <0.001 |
| Non-melanoma skin cancer | 17 | 0.90 (0.60 to 1.34) | 0.60 | <0.001 |
| Lung cancer | 20 | 0.62 (0.51 to 0.75) | <0.001 | <0.001 |
| Colorectal cancer | 20 | 0.82 (0.75 to 0.90) | <0.001 | <0.001 |
| Breast cancer | 15 | 1.02 (0.93 to 1.12) | 0.66 | 0.001 |
| Prostate cancer | 17 | 0.93 (0.83 to 1.03) | 0.18 | <0.001 |

*Smoking-related cancer includes cancer of the lung, larynx, mouth, oesophagus, throat, bladder, kidney, liver, stomach, pancreas, colon and rectum, and cervix, as well as acute myeloid leukaemia.
†Non-smoking-related cancer includes all other cancer except for those listed as smoking related.
RR, relative risk.;

a high risk of many types of cancer.[7] Moreover, there is evidence that PD patients are less likely to be smokers.[79] Of note, only 10 out of the 32 publications included were adjusted for smoking behaviour for total cancer risk in their original analysis,[18 25 27–31 51 64 75] which may introduce residual confounding for the observed association between PD and total cancer. However, we also found that non-smoking-related cancer was inversely associated with PD, even when melanoma was included. Because only four publications separately reported risk

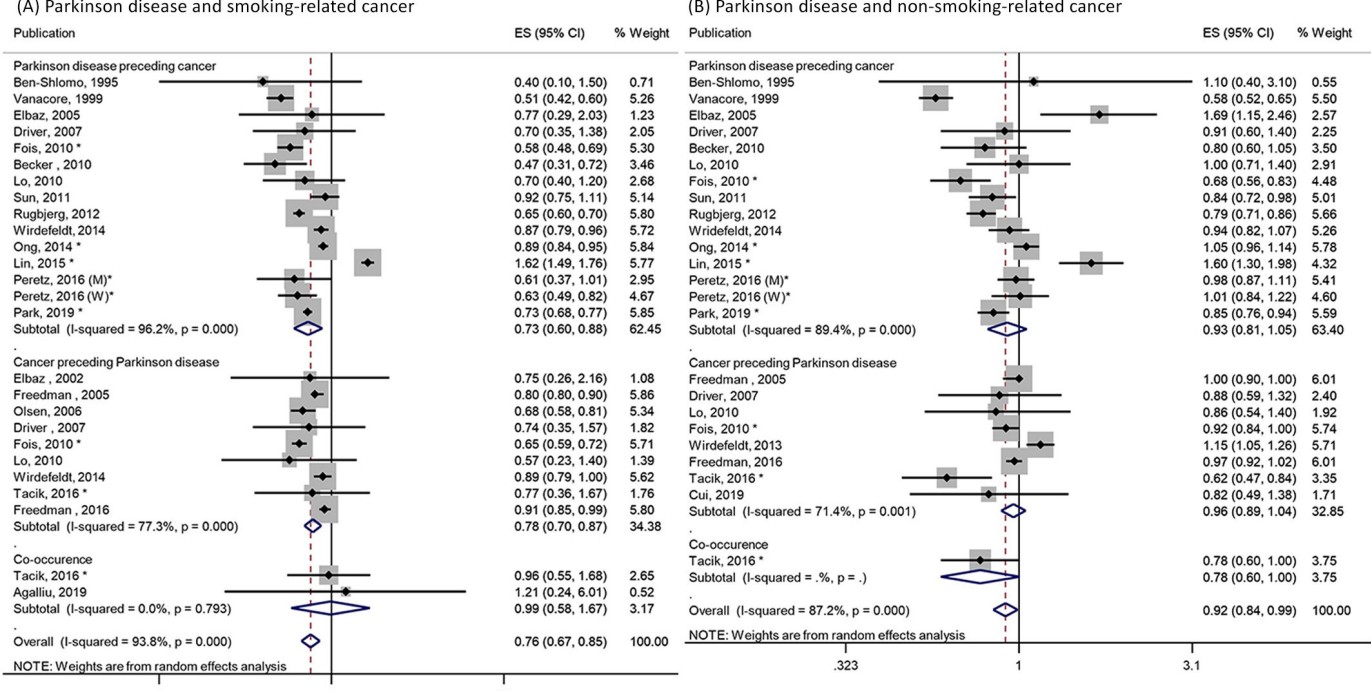

**Figure 3** Association between Parkinson's disease and (A) smoking-related cancers in 21 publications and (B) non-smoking-related cancers in 19 publications. Figure shows the estimates (ESs) and 95% CIs for each study and the pooled result from random effects model. Studies are stratified by temporal relationship of Parkinson's disease and cancer. *Pooled risk estimates calculated from individual ES in original publication. M, men; W, women.

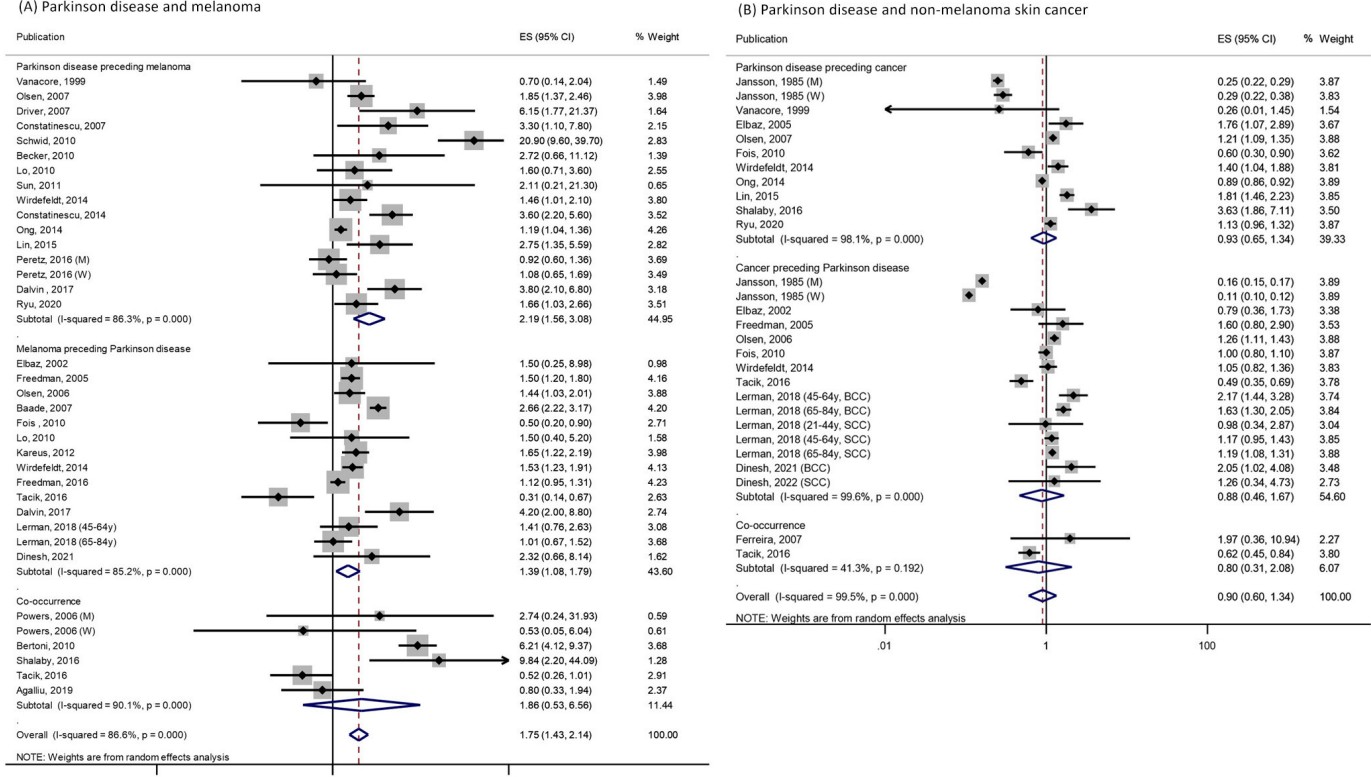

**Figure 4** Association between Parkinson's disease and (A) melanoma in 29 publications and (B) non-melanoma skin cancers in 17 publications. Figure shows the estimates (ESs) and 95% CIs for each study and the pooled result from random effects model. Studies are stratified by temporal relationship of Parkinson's disease and cancer. BCC, basal cell carcinoma; M, men; SCC, squamous cell carcinoma; W, women; Y, years of age.

estimates for total cancer or non-smoking-related cancers after excluding melanoma,[25 54 55 76] we did not perform a meta-analysis in these secondary categories. Our findings suggest that smoking is unlikely the only factor contributing to the observed inverse relation between PD and total cancer. Future studies with carefully adjusted smoking habits or environmental smoking exposure are warranted to better address this issue.

Our results, in line with the previous meta-analysis,[10 78] suggest an inverse comorbidity between PD and cancer. The biological bases underlying the association is far from clear. Dysregulated cellular processes including those involved in the regulation of cell cycle, mitochondrial function, DNA repair, cell metabolism and immune responses have been implicated in degeneration of neurons and tumourigenesis in dividing cells, often in the opposite directions. Cell proliferation and survival signals such as Wnt, P53, and PI3K/AKT may be upregulated in cancer and downregulated in neurodegeneration. The ubiquitin proteasome pathway of protein degradation on the other hand may be downregulated in neurodegeneration and upregulated in cancer.[80–82] Understanding the biological pathways would further facilitate investigations on potential strategies for better prevention, surveillance and treatment of both PD and cancer.

Several common gene mutations have been implicated in PD and cancer.[83] *PARK2* was found to be a potent tumour suppressor gene.[84 85] Other PD-related genes

*PINK1, PARK7* and *LRRK2* have also been linked to cancer.[8 60 86] PD patients carrying *LRRK2* G2019S mutation have been associated with an overall increased risk of cancer, especially for hormone-related cancer and breast cancer,[84] and most recently, lukaemia, colon cancer, and skin cancer when compared with noncarrier PD.[87] Another PD-related *LRRK2* mutation R1441G was found to be associated with higher prevalence of haematological cancers.[88] Both G2019S and R1441G show increased LRRK2 kinase activity.[89] However, a recent study demonstrated that loss of *LRRK2* could promote lung cancer development, adding to the complexicity of LRRK2-cancer link.[90] We found that only 14 of the included studies specifically identified idiopathic PD and excluded genetically determined PD. This limits our systematic review to distinguish genetic forms of PD from idiopathic PD and fully synthesise the potential genetic overlaps between PD and cancer.

Although similarly characterised pathologically by over proliferation, different cancers are highly heterogeneous. While it remains to be determined whether the general inverse association exists across cancers of different sites and evolutionary origins, we and others have consistently shown that it did not apply to melanoma.[9 91] In this meta-analysis, we replicated the well-documented positive link between PD and melanoma. It has long been proposed that levodopa as the mainstay therapy for PD and a common precursor for both dopamine and melanin may

contribute to the higher risk of melanoma in PD.[92 93] In this meta-analysis, we found a 37% higher risk of newly developed PD after diagnosis of melanoma, suggesting that the observed PD-melanoma association may not be fully explained by the role of levodopa, if any.[94] Previously, we reported that the risk of incident PD is higher in people with a family history of melanoma among their first-degree relatives.[91] One plausible biological explanation of the association is the regulation of pigmentation by the *MC1R* gene, which presents and functions in both melanocytes and dopaminergic neurons.[95 96] Other genetic mutations, such as *CYP2D6* polymorphism and *VDR* polymorphism, might also be involved in both conditions.[97–100]

Despite all our effort in synthesising all epidemiological evidence, the intrinsic limitations of meta-analysis cannot be avoided. First, studies included in this analysis came from diverse populations, with diverse designs and treatment strategies. They varied across assessments, statistical methods, and adjusted covariates. Although meta-regression did not find differences in age, sex, ethnicity, study design and study quality, the highly heterogeneous nature of this meta-analysis limits its interpretation into robust conclusions. Second, due to lack of access to original data, we could not adjust uniformly for confounders. We addressed this shortcoming by stratifying cancers into smoking-related or non-smoking-related cancers. However, there may be residual confounding since only a few studies adjusted for family history of PD/cancer, use of medications, sun exposure, duration of PD/cancer, use of medical care or diet (eg, caffeine consumption).[101–103] Third, many large-scale studies included in this meta-analysis used local/national registry databases, with disease diagnosis mostly based on International Classification of Disease codes. Notification to registries might not be complete, therefore the cases might be under-reported. Moreover, diagnosis criteria may slightly vary in different countries, hospitals, etc. Thus it is challenging to confirm and validate the information from these datasets. Lastly, all publications included in this meta-analysis were based on populations from North America, Europe, Australia and Central and East Asia; No study has examined the association of PD and cancer in less-developed regions such as Africa, Southeast Asia or South America. This could be due to difficulties in disease diagnosis and registry in these regions. Recent findings suggested positive associations between PD and most cancers in an East Asian population, highlighting possible discrepancies among different populations with different ethnic backgrounds.[8 50] Future studies should address the potentially important role of race/ethnicity and socialeconomic status.

We reviewed the current epidemiological evidence for the association between cancer and PD, with a meta-analysis of over 17 million individuals. We found that PD was associated with low risk of total cancer, except for melanoma, with which a positive association was identified. Despite the limitations, our study provided an overall picture of the association between the two major disease entities. Future studies should aim to better understand the links between these two major chronic disease entities using epidemiological, clinical and biological approaches.

**Correction notice** Since this article was first published online the author name Xinyuan has been corrected to Xinyuan.

**Acknowledgements** We thank Christina Wissinger, PhD, from Health Sciences Liaison Librarian of Penn State, who helped with search strategy. No financial compensation was made.

**Contributors** XZ, The Pennsylvania State University: concept and design; acquisition, analysis or interpretation of data; statistical analysis; drafting of the manuscript; revised the manuscript for intellectual content. DG, Massachusetts General Hospital: concept and design; acquisition, analysis or interpretation of data; revised the manuscript for intellectual content. NM, Massachusetts General Hospital: acquisition, analysis or interpretation of data; revised the manuscript for intellectual content. XC, Massachusetts General Hospital: concept and design; acquisition, analysis or interpretation of data; drafting of the manuscript; revised the manuscript for intellectual content. XG, The Pennsylvania State University: concept and design; acquisition, analysis or interpretation of data; drafting of the manuscript; revised the manuscript for intellectual content.

**Funding** This work was supported by the National Institutes of Health (R01NS102735), the Parkinson's Disease Foundation (PF-APDA-SFW-1914), the Jane & Alan Batkin Foundation (N/A), the Farmer Family Foundation (N/A) and the Michael J. Fox Foundation and the Aligning Science Across Parkinson's Initiative (ASAP-000312).

**Disclaimer** None of the listed funders had a role in the design and conduct of the study; collection, management, analysis, and interpretation of the data; preparation, review, or approval of the manuscript; and decision to submit the manuscript for publication.

**Competing interests** None declared.

**Patient and public involvement statement** It was not appropriate to involve patients or the public in the design, or conduct, or reporting, or dissemination plans of our research.

**Provenance and peer review** Not commissioned; externally peer reviewed.

**Data availability statement** All data relevant to the study are included in the article or uploaded as online supplemental information. The data used to support the findings of this article are included within the article and online supplemental material.

**ORCID iDs**
David Guarin http://orcid.org/0000-0003-3011-3252
Xiang Gao http://orcid.org/0000-0003-2617-6509

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
