## [Reviewer comments · BMJ Open]

ARTICLE DETAILS

TITLE (PROVISIONAL)	Parkinson's disease and cancer: a systematic review and meta-analysis of over 17 million participants
AUTHORS	Zhang, Xinyaun; Guarin, David; Mohammadzadehhonarvar, Niyaz; Chen, Xiqun; Gao, Xiang

VERSION 1 – REVIEW

REVIEWER	Natalia Madetko Medical University of Warsaw, Neurology
REVIEW RETURNED	12-Jan-2021

GENERAL COMMENTS	Zhang et al. in a paper titled "Parkinson's disease and cancer: a systematic review and meta-analysis of 17,697,252 participants" presented epidemiological evidence on associations between Parkinson's disease and cancer. In a meta-analysis including 61 studies and 17 697 252 participants, Authors found inverse association between PD and both smoking- and non-smoking-related cancers and positive association with melanoma. Meta-analysis included assessment of possible associations in many aspects : PD preceding cancer, cancer preceding PD, co-occurrence, presence of levodopa treatment, division into smoking-related and non-smoking related cancers, specific analysis for melanoma, non-melanoma skin cancers and other cancers. Main limitations I find are: 1. ABSTRACT : Does not include introduction describing significance and implications of possible correlations between PD and cancer.2. INTRODUCTION : In my opinion, this section should be enriched by adding paragraph considering genetic, inflammatory and environmental background of PD, which might be shared with some neoplasms.3. METHODOLOGY : Due to some genetic overlaps between PD and neoplasms, it would be beneficial to assess, if possible, correlations included in the paper separately for idiopathic and genetically conditioned Parkinson's disease.4. DISCUSSION : The manuscript would benefit if more recent references were added. However, including more original papers could jeopardize the feasibility of obtained conclusions. In this context adding not necessarily original studies published within the last year as point of view in the discussion could highlight the progress of the issue to the readership of the journal. Consider, please, if the following papers are worthy to be quoted:
---

	Dube U, Ibanez L, Budde JP, Benitez BA, Davis AA, Harari O, Iles MM, Law MH, Brown KM; 23andMe Research Team; Melanoma-Meta-analysis Consortium, Cruchaga C. Overlapping genetic architecture between Parkinson disease and melanoma. Acta Neuropathol. 2020 Feb;139(2):347-364. doi: 10.1007/s00401-019-02110-z. Epub 2019 Dec 16. Erratum in: Acta Neuropathol. 2020 Mar 14;: PMID: 31845298; PMCID: PMC7379325. Ejma M, Madetko N, Brzecka A, Guranski K, Alster P, Misiuk-Hojło M, Somasundaram SG, Kirkland CE, Aliev G. The Links between Parkinson's Disease and Cancer. Biomedicines. 2020 Oct 14;8(10):416. doi: 10.3390/biomedicines8100416. PMID: 33066407; PMCID: PMC7602272. Weissenrieder JS, Neighbors JD, Mailman RB, Hohl RJ. Cancer and the Dopamine D2 Receptor: A Pharmacological Perspective. J Pharmacol Exp Ther. 2019 Jul;370(1):111-126. doi: 10.1124/jpet.119.256818. Epub 2019 Apr 18. PMID: 31000578; PMCID: PMC6558950. General comment: It is a valuable work, which assess epidemiological correlations between PD and neoplasms. The evident strengths of this study are:  • the large number of analyzed participants • statistical efforts to reduce the impact of heterogeneity between included papers • critical point of view considering obtained results The major limitations I find are listed above. Additionally, Authors should extend their references to current literature – presently only about 30% of references were published during the last 5 years. The suggested references are only some, which could be implemented in the text. The work should be re-evaluated after minor revision.
--	--

REVIEWER	Luca Marsili University of Cincinnati, Neurology and Rehabilitation Medicine
REVIEW RETURNED	29-Jan-2021

GENERAL COMMENTS	In this manuscript, the authors have done a big data analysis of all epidemiological evidence on associations between Parkinson disease (PD) and cancer via meta-analysis. They found, as expected, that PD and total cancer were inversely associated. This inverse association persisted for both smoking-related and non-smoking-related cancers. In contrast, PD was positively associated with melanoma. The present manuscript has the merit of investigating the important question of the association between neurodegeneration and cancer, using a meta-analytic approach for big data analysis. However, some aspects must be improved:  - Title: I would avoid writing the digits "17,697,252" because it is confusing, and would rather write something like "a big data analysis" (or something similar). - Abstract: please, spell "PD" entirely for the first time you mention the abbreviation. - Article Summary: Please, use the plural term "meta-analyses"
---

	- Introduction: 1) I believe that some clarifications can be done here. For instance, clear distinction has to be made between sporadic (non-Mendelian) PD and genetically-determined Parkinsonisms throughout the text. The authors then mention that "Growing epidemiological evidence suggests that PD and cancer may be inversely associated". I would specify here that some recent studies have deeply investigated this relationship concluding that: a) skin cancer may have an effect on delaying the onset but not the progression of idiopathic PD; b) genetic PD and cancer may have common pathways. These two aspects have to be discussed in the introduction and / or discussion sections. 2) The authors then mention that "PD and cancers are both rare diseases." I do not agree with this statement. PD is the second most frequent neurodegenerative disorder and cancer incidence increases with age, so it is highly dependent on the population age of interest, that in PD is the geriatric one (so much more frequent than in young adults). 3) The authors then discuss the temporal association between PD and cancer: PD preceding cancer, and cancer preceding PD. Did they use any cutoff? It is well known that PD before manifesting clinically takes several years. Some authors have included a cutoff of at least 2 or 5 years. Please, clarify. - Methods: "The inclusion criteria were: studies.. that was" please change with "were". Also, in point 3), please clarify if genetic Parkinsonisms and/or atypical parkinsonian syndromes were excluded. Among Exclusion criteria the authors mention "reviews", but what about other previous meta-analyses? How did they handle this issue? Please, clarify. Later they mention that "two first authors..screened the publications." Please, add their qualifications. Moreover, it is not clear to me if any quality appraisal scoring was done. Did the authors use any quality scoring to weight the meta-analysis results? Also, the discussion needs to incorporate some limitations of different studies of how measurements of outcomes and ascertainment of cancer was done. Is this included in the quality rating assessment (if done)? - Data extraction: Please, clarify the meaning of "dominant sex" - Statistical analysis: Page 8, line 53: "precedings/abstract." Maybe the authors intended "Proceedings"? Please, clarify. - Results, "Other site specific cancers": "There was no association between PD and breast and prostate cancer." Is this result valid after correction for age? Elderly patients usually have both PD and prostate cancers, please add a comment in the discussion section. - Discussion: when mentioning LRRK2 PD-related gene, please discuss the different findings among the LRRK2 mutations: G2019S and R441G; it looks like they are associated with different cancers, thus underlying the concept that specific altered pathways might be associated with different cancers. Also, please write all gene names in italics. Finally, recent meta-analyses have investigated the interplay between genetic parkinsonism and cancer, and this aspect could be mentioned in this section of the discussion. Additionally, when discussing the confounders (page 14, lines 52-55), I would also mention the sun-exposure as an important factor contributing to skin cancers (see also Mahajan A, et al 2020. Skin Cancer May Delay Onset but Not Progression of Parkinson's Disease: A Nested Case-Control Study. Front. Neurol. 11:406. doi: 10.3389/fneur.2020.00406). - Supplementary methods, page 33, line 5, please change with "proceedings" - Supplementary table 1: Is the first study Agalliu 2019? Please
--	---

	clarify. Also, in the table headings, please clarify the meaning of "direction." How was defined cancer preceding PD or vice versa? Also what does it mean "diagnosis"? How was diagnosis achieved? Histologically-based? please, clarify.
REVIEWER	Catherine Klersy Fondazione IRCCS Policlinico San Matteo, Service of Clinical Epidemiology & Biometry, Research Department
REVIEW RETURNED	17-Mar-2021
GENERAL COMMENTS	The analysis is well conducted, using an appropriate statistical methodology, with current guidelines followed. Particularly appreciated is the attention on potential confounding by using meta regression and the subgroup analysis to conform results. Figures are clear. The discussion and conclusions are well supported by the data, with the epidemiological limitations well underlined

VERSION 1 – AUTHOR RESPONSE

Reviewer: 1

Dr. Natalia Madetko, Medical University of Warsaw

Comments to the Author:

Zhang et al. in a paper titled “Parkinson’s disease and cancer: a systematic review and meta-analysis of 17,697,252 participants” presented epidemiological evidence on associations between Parkinson’s disease and cancer. In a meta-analysis including 61 studies and 17 697 252 participants, Authors found inverse association between PD and both smoking- and non-smoking-related cancers and positive association with melanoma. Meta-analysis included assessment of possible associations in many aspects : PD preceding cancer, cancer preceding PD, co-occurrence, presence of levodopa treatment, division into smoking-related and non-smoking related cancers, specific analysis for melanoma, non-melanoma skin cancers and other cancers.

Main limitations I find are:

1. **ABSTRACT** : Does not include introduction describing significance and implications of possible correlations between PD and cancer.

Response: With the word limit, we have added a short sentence in conclusions to indicate broad implication of our findings:

“These results provide evidence for further investigations for possible mechanistic associations between PD and cancer.”

2. **INTRODUCTION** : In my opinion, this section should be enriched by adding paragraph considering genetic, inflammatory and environmental background of PD, which might be shared with some neoplasms.

Response: We have revised this section accordingly.

3. **METHODOLOGY** : Due to some genetic overlaps between PD and neoplasms, it would be beneficial to assess, if possible, correlations included in the paper separately for idiopathic and genetically conditioned Parkinson’s disease.

Response: We examined the included studies and found that only 14 of them specifically identified the PD cases as idiopathic PD. These publications were identified in the Supplementary table 1. We have added this limitation in the manuscript:

“We found that only 14 of the included studies specifically identified idiopathic PD and excluded genetically conditioned PD. This limits our systematic review to fully synthesize the potential genetic overlaps between PD and cancer.”

4. DISCUSSION : The manuscript would benefit if more recent references were added. However, including more original papers could jeopardize the feasibility of obtained conclusions. In this context adding not necessarily original studies published within the last year as point of view in the discussion could highlight the progress of the issue to the readership of the journal. Consider, please, if the following papers are worthy to be quoted:

Dube U, Ibanez L, Budde JP, Benitez BA, Davis AA, Harari O, Iles MM, Law MH, Brown KM; 23andMe Research Team; Melanoma-Meta-analysis Consortium, Cruchaga C. Overlapping genetic architecture between Parkinson disease and melanoma. *Acta Neuropathol.* 2020 Feb;139(2):347-364. doi: 10.1007/s00401-019-02110-z. Epub 2019 Dec 16. Erratum in: *Acta Neuropathol.* 2020 Mar 14;: PMID: 31845298; PMCID: PMC7379325.

Ejma M, Madetko N, Brzecka A, Guranski K, Alster P, Misiuk-Hojło M, Somasundaram SG, Kirkland CE, Aliev G. The Links between Parkinson's Disease and Cancer. *Biomedicines.* 2020 Oct 14;8(10):416. doi: 10.3390/biomedicines8100416. PMID: 33066407; PMCID: PMC7602272.

Weissenrieder JS, Neighbors JD, Mailman RB, Hohl RJ. Cancer and the Dopamine D2 Receptor: A Pharmacological Perspective. *J Pharmacol Exp Ther.* 2019 Jul;370(1):111-126. doi: 10.1124/jpet.119.256818. Epub 2019 Apr 18. PMID: 31000578; PMCID: PMC6558950.

Response: We have revised Discussion and updated references to include suggested publications (ref 84,5,95) and other newer publications.

General comment:

It is a valuable work, which assess epidemiological correlations between PD and neoplasms.

The evident strengths of this study are:

- the large number of analyzed participants
- statistical efforts to reduce the impact of heterogeneity between included papers
- critical point of view considering obtained results

The major limitations I find are listed above. Additionally, Authors should extend their references to current literature – presently only about 30% of references were published during the last 5 years.

The suggested references are only some, which could be implemented in the text.

The work should be re-evaluated after minor revision.

Response: We have included more recent literature in the revision. We have also updated our search for relevant literature and meta-analysis up until March, 2021.

Reviewer: 2

Dr. Luca Marsili, University of Cincinnati

Comments to the Author:

In this manuscript, the authors have done a big data analysis of all epidemiological evidence

on associations between Parkinson disease (PD) and cancer via meta-analysis. They found, as expected, that PD and total cancer were inversely associated. This inverse association persisted for both smoking-related and non-smoking-related cancers. In contrast, PD was positively associated with melanoma.

The present manuscript has the merit of investigating the important question of the association between neurodegeneration and cancer, using a meta-analytic approach for big data analysis. However, some aspects must be improved:

- Title: I would avoid writing the digits "17,697,252" because it is confusing, and would rather write something like "a big data analysis" (or something similar).

Response: We have revised the title.

- Abstract: please, spell "PD" entirely for the first time you mention the abbreviation.

Response: We have revised accordingly.

- Article Summary: Please, use the plural term "meta-analyses"

Response: We have revised accordingly.

- Introduction:

1) I believe that some clarifications can be done here. For instance, clear distinction has to be made between sporadic (non-Mendelian) PD and genetically-determined Parkinsonisms throughout the text. The authors then mention that "Growing epidemiological evidence suggests that PD and cancer may be inversely associated". I would specify here that some recent studies have deeply investigated this relationship concluding that: a) skin cancer may have an effect on delaying the onset but not the progression of idiopathic PD; b) genetic PD and cancer may have common pathways. These two aspects have to be discussed in the introduction and / or discussion sections.

Response: In this meta-analysis, we found that only 14 of the included studies specifically identified idiopathic PD and excluded genetically conditioned PD. This limits our systematic review to fully synthesize the potential genetic overlaps between PD and cancer. Nevertheless, we have revised Discussion and added the recent findings.

2) The authors then mention that "PD and cancers are both rare diseases." I do not agree with this statement. PD is the second most frequent neurodegenerative disorder and cancer incidence increases with age, so it is highly dependent on the population age of interest, that in PD is the geriatric one (so much more frequent than in young adults).

Response: We have removed the sentence from the manuscript

3) The authors then discuss the temporal association between PD and cancer: PD preceding

cancer, and cancer preceding PD. Did they use any cutoff? It is well known that PD before manifesting clinically takes several years. Some authors have included a cutoff of at least 2 or 5 years. Please, clarify.

Response: The temporal association was defined by study design and based on the diagnosis and cut-offs in each individual study. The majority of studies used the diagnosis date as the index date for both diseases. Moreover, the cohort studies differed in whether there is a lag time for follow-up (from no lag to 5-year lag). We have addressed this question in the manuscript:

"The temporal association was defined per each individual study definition, most of which was based on the diagnosis date of the two diseases."

- Methods: "The inclusion criteria were: studies that was" please change with "were". Also, in point 3), please clarify if genetic Parkinsonisms and/or atypical parkinsonian syndromes were excluded. Among Exclusion criteria the authors mention "reviews", but what about other previous meta-analyses? How did they handle this issue? Please, clarify. Later they mention that "two first authors..screened the publications." Please, add their qualifications.

Response: Only 14 of the included studies specifically identified the PD cases as idiopathic PD. We have addressed this the discussion section:

"We found that only 14 of the included studies specifically identified idiopathic PD and excluded genetically conditioned PD. This limits our systematic review to fully synthesize the potential genetic overlaps between PD and cancer."

Our study excluded any results that only used parkinsonism patients that did not meet the criteria for PD. This is specified in the manuscript:

"Parkinsonism that does not meet the criteria for PD and benign neoplasm were not included."

We excluded previous systematic reviews and meta-analyses, but used them as references for manual searching of related publications. This has been clarified in the manuscript:

"Previous meta-analyses were used as references for manual searching of related publications."

The credentials of the two first authors have been added:

"Two first authors (X. Z., BS and D. G., BA) ..."

Moreover, it is not clear to me if any quality appraisal scoring was done. Did the authors use any quality scoring to weight the meta-analysis results? Also, the discussion needs to incorporate some limitations of different studies of how measurements of outcomes and ascertainment of cancer was done. Is this included in the quality rating assessment (if done)?

Response: We examined the quality of included studies following the Newcastle-Ottawa Scale for cohort and case-controls studies, and the scores assigned to each publication were added in Supplementary table 1. We have detailed the assessment in Methods:

“Study quality was assessed by the Newcastle-Ottawa Scale for cohort studies and for case-control studies¹³, which is based on the definition of case/control, the definition of exposure/outcome, covariates, and other relevant factors. The score ranged from 0–9, and we separated the included studies into low quality group (< 7) and high quality group (≥ 7), based on the mean quality score of the included studies. Proceedings/abstracts were not included in the quality check.”

- Data extraction: Please, clarify the meaning of "dominant sex"

Response: We have specified in the Methods:

“Dominant sex and ethnicity were defined as the major sex and race/ethnicity (>50%) of the studied population, respectively.”

- Statistical analysis: Page 8, line 53: "precedings/abstract." Maybe the authors intended "Proceedings"? Please, clarify.

Response: we have corrected the word into “proceedings”.

- Results, "Other site specific cancers": "There was no association between PD and breast and prostate cancer." Is this result valid after correction for age? Elderly patients usually have both PD and prostate cancers, please add a comment in the discussion section.

Response: The majority of included studies controlled for age in their original risk estimations.

- Discussion: when mentioning LRRK2 PD-related gene, please discuss the different findings among the LRRK2 mutations: G2019S and R441G; it looks like they are associated with different cancers, thus underlying the concept that specific altered pathways might be associated with different cancers. Also, please write all gene names in italics. Finally, recent meta-analyses have investigated the interplay between genetic parkinsonism and cancer, and this aspect could be mentioned in this section of the discussion.

Response: We have revised the discussion accordingly. However, given that only 14 of the included studies specifically identified the PD cases as idiopathic PD and excluded genetically conditioned PD. We were not able to fully address the potential genetic overlaps between PD and cancer (see related responses above).

Additionally, when discussing the confounders (page 14, lines 52-55), I would also mention the sun-exposure as an important factor contributing to skin cancers (see also Mahajan A, et al

2020. Skin Cancer May Delay Onset but Not Progression of Parkinson’s Disease: A Nested Case-Control Study. Front. Neurol. 11:406. doi: 10.3389/fneur.2020.00406).

Response: We have revised the discussion accordingly, as below:

“However, there may be residual confounding since only a few studies adjusted for family history of PD/cancer, use of medications, sun exposure, duration of PD/cancer, ...”

- Supplementary methods, page 33, line 5, please change with "proceedings"

Response: We have revised accordingly.

- Supplementary table 1: Is the first study Agalliu 2019? Please clarify. Also, in the table headings, please clarify the meaning of "direction." How was defined cancer preceding PD or vice versa? Also what does it mean "diagnosis"? How was diagnosis achieved? Histologically-based? please, clarify.

Response: We have revised Supplementary Table 1. First, we have added two new publications from the updated literature search in March 2020 – March 2021 (Ryu 2020, Dinesh 2021). Second, “Direction” in the headline has been changed to “Temporal direction “.

Third. “Diagnosis” in the headline has been changed to “Disease ascertainment”. Footnotes have been added to the table in clarifying these questions:

“Study design and temporal direction was defined per each individual study definition, most of which was based on the diagnosis date of two diseases. Disease ascertainment was defined per the description of whether any physicians, neurologists or movement specialists made the diagnosis. Quality score was assessed by the Newcastle-Ottawa Scale for cohort studies and for case-control studies (range 0–9).” (Supplementary material page 9)

Reviewer: 3

Dr. Catherine Klersy, Fondazione IRCCS Policlinico San Matteo

Comments to the Author:

The analysis is well conducted, using an appropriate statistical methodology, with current guidelines followed. Particularly appreciated is the attention on potential confounding by using meta regression and the subgroup analysis to conform results. Figures are clear. The discussion and conclusions are well supported by the data, with the epidemiological limitations well underlined

Response: We thank the reviewer for this comment.

VERSION 2 – REVIEW

REVIEWER	Natalia Madetko Medical University of Warsaw, Neurology
REVIEW RETURNED	18-May-2021
GENERAL COMMENTS	I endorse the publication of this manuscript in its current form.

REVIEWER	Luca Marsili University of Cincinnati, Neurology and Rehabilitation Medicine
REVIEW RETURNED	13-May-2021

GENERAL COMMENTS	I commend the authors for their great work. I have only a few minor comments:  - Introduction page 5, line 27: Please, change "mitochodia" with "mitochondria" - Page 7, line 34: Please add "idiopathic" when discussing the criteria for PD. In general, I would use the word idiopathic to differentiate from genetically determined PD. - Page 14, line 43: Please, change "genetically conditioned PD" with "genetically determined PD"
--

VERSION 2 – AUTHOR RESPONSE

Reviewer: 2

Dr. Luca Marsili, University of Cincinnati Comments to the Author:

I commend the authors for their great work.

I have only a few minor comments:

- Introduction page 5, line 27: Please, change "mitochodia" with "mitochondria"

Response: We have made the change

- Page 7, line 34: Please add "idiopathic" when discussing the criteria for PD. In general, I would use the word idiopathic to differentiate from genetically determined PD.

Response: Given that most of the studies included in our analyses did not distinguish genetic PD from idiopathic PD, we agree with the reviewer that the PD cases covered by this meta-analysis were most likely but not exclusively idiopathic PD. We revised the sentence in Discussion on page 14 as below:

"We found that only 14 of the included studies specifically identified idiopathic PD and excluded genetically determined PD. This limits our systematic review to distinguish genetic forms of PD from idiopathic PD and fully synthesize the potential genetic overlaps between PD and cancer"

- Page 14, line 43: Please, change "genetically conditioned PD" with "genetically determined PD"

Response: We have changed the wording

Reviewer: 1

Dr. Natalia Madetko, Medical University of Warsaw Comments to the Author:

I endorse the publication of this manuscript in its current form.

On behalf of all authors, I thank you and the reviewers for your time and consideration. We look forward to your final decision.